# VALUE: A Multi-Task Benchmark for Video-and-Language Understanding Evaluation

**Linjie Li**[*1], **Jie Lei**[*2], **Zhe Gan**[1], **Licheng Yu**[2], **Yen-Chun Chen**[1],
**Rohit Pillai**[1], **Yu Cheng**[1], **Luowei Zhou**[1], **Xin Eric Wang**[3], **William Yang Wang**[4],
**Tamara L. Berg**[2], **Mohit Bansal**[2], **Jingjing Liu**[5], **Lijuan Wang**[1], **Zicheng Liu**[1]

[1]Microsoft Corporation    [2]UNC Chapel Hill
[3]UC Santa Cruz    [4]UC Santa Barbara    [5]Tsinghua University

```
{lindsey.li,zhe.gan,yen-chun.chen,rohit.pillai,
yu.cheng,luowei.zhou,lijuanw,zliu}@microsoft.com
{jielei,licheng,tlberg,mbansal}@cs.unc.edu
xwang366@ucsc.edu, william@cs.ucsb.edu, JJLiu@air.tsinghua.edu.cn
```

## Abstract

Most existing video-and-language (VidL) research focuses on a single dataset, or multiple datasets of a single task. In reality, a truly useful VidL system is expected to be easily generalizable to diverse tasks, domains, and datasets. To facilitate the evaluation of such systems, we introduce **V**ideo-**A**nd-**L**anguage **U**nderstanding **E**valuation (**VALUE**) benchmark, an assemblage of 11 VidL datasets over 3 popular tasks: ($i$) text-to-video retrieval; ($ii$) video question answering; and ($iii$) video captioning. VALUE benchmark aims to cover a broad range of video genres, video lengths, data volumes, and task difficulty levels. Rather than focusing on single-channel videos with visual information only, VALUE promotes models that leverage information from both video frames and their associated subtitles, as well as models that share knowledge across multiple tasks. We evaluate various baseline methods with and without large-scale VidL pre-training, and systematically investigate the impact of video input channels, fusion methods, and different video representations. We also study the transferability between tasks and conduct multi-task learning under different settings. The significant gap between our best model and human performance calls for future study for advanced VidL models. VALUE is available at https://value-benchmark.github.io/.[2]

## 1 Introduction

Joint video-and-language (VidL) understanding sits at the nexus of computer vision and natural language processing (NLP), and has attracted rapidly growing attention from both communities. Popular tasks include text-based video retrieval [70, 28, 53, 35, 36], video moment retrieval [7, 21, 28, 35, 36], video question answering [68, 25, 32, 33], and video captioning [53, 70, 74, 35]. However, existing works [41, 26, 29, 20, 50, 44] in this field are often evaluated on distinct datasets under different experimental settings, making fair comparison difficult between methods. Meanwhile, most works are evaluated on a limited set of tasks, thus difficult to measure as a universal VidL system. As exemplary, in the NLP community, GLUE [62] and SuperGLUE [61] have evolved into prominent evaluation frameworks that continue to push the frontier of natural language understanding, due to their broad coverage of NLP tasks with diverse training data volumes, task genres, and unified task formulation.

---

[*] Equal contribution.

[2]VALUE competition will be held in conjunction with CLVL workshop at ICCV 2021, https://sites.google.com/view/iccv21clvl/home.

Table 1: **Statistics of video data** used in VALUE benchmark. Multi-channel ratio refers to percentage of videos with subtitles. Video lengths are measured in terms of seconds (s) on average.

| Video Data | Source | #Video | Multi-channel Ratio | Length |
|---|---|---|---|---|
| TV (TVQA, TVR, TVC) | TV episodes | 21.8K | 100% | 76s |
| How2 (How2R, How2QA) | Instructional Videos on Youtube | 31.7K | 99.36% | 59s |
| VIOLIN | TV episodes, Movie Clips | 15.9K | 99.33% | 40s |
| VLEP | TV episodes, Vlog on Youtube | 10.2K | 98.11% | 32s |
| YouCook2 (YC2C, YC2R) | Cooking Videos on Youtube | 15.4K | 94.40% | 20s |
| VATEX (VATEX-EN-R/C) | Various Youtube Videos | 41.3K | 50.93% | 10s |

Inspired by them, to better benchmark advances in VidL research, we introduce **V**ideo-**A**nd-**L**anguage **U**nderstanding **E**valuation (**VALUE**) benchmark, an online platform with a compilation of 11 VidL datasets for model evaluation and comparison. There are several contributions that render VALUE a unique and valuable asset to the community. ($i$) **Diversity**: To evaluate the versatility and generalizability of VidL systems, our benchmark includes diverse tasks, including video retrieval, question answering (QA), and captioning (see Section 3 for details). VALUE also covers a broad range of video genres, video lengths, and data volumes. ($ii$) **Multi-channel video inputs**: Videos are multi-channeled and usually contain frames, audio, and textual information. Most of the existing works, however, only focus on the use of video frames. In our benchmark, we provide both video frames and their accompanying dialogues in the form of subtitle sentences[3] as video inputs. Tasks that require multi-channel information for inference are preferable. In TVQA [32], for example, the cues to answering the questions are usually in both visual and dialogue content. ($iii$) **Task difficulties**: Our benchmark is challenging and hard-to-game. We found that even the best VidL models we tested underperform human baselines by a large margin, suggesting great space for improvement. ($iv$) **Easy evaluation**: For each dataset, we select a representative metric from a set of standard metrics for evaluation. We divide the datasets into 3 categories, and rank participants in each category based on the meta-average score across associated tasks. For the VALUE leaderboard, we provide a universal target metric (*i.e.*, the meta-average score across all the tasks) to track progress. We also release rich pre-extracted video frame features, offer starter code, and withhold private test data for reliable evaluation on our evaluation server.

To provide an in-depth analysis of our VALUE benchmark, we evaluate a number of baselines with and without pre-training, and systematically assess the effects of video input channels, fusion methods, and different video representations. We also investigate the transferability between tasks and the effect of multi-task training under various settings (*e.g.*, multi-task learning by task type or by data domain). Video-and-language understanding is challenging, as it encompasses a wide range of areas such as visual and linguistic semantic understanding, spatial-temporal grounding, multimodal fusion, and commonsense reasoning. We envision that VALUE will inspire active research and discussion in the community. More details are available at `https://value-benchmark.github.io/`.

## 2 Related Work

Publicly accessible large-scale multi-task benchmarks [12, 62, 61, 23, 37] have facilitated recent advances [16, 72, 42, 71, 64, 18, 47] in NLP. For example, SentEval [12] contains a collection of natural language tasks, such as sentiment analysis [48, 56], entailment [10] and semantic textual similarity [4, 5, 2, 1, 3]. While SentEval aims at evaluating sentence-level vector representations, GLUE [62] advanced it by removing all restrictions on the model – GLUE is designed to be model-agnostic, allowing the evaluation of any type of representation. With the introduction of large-scale transformer [59] language models such as BERT [16], RoBERTa [42], XLNET [72] and OpenAI GPT [52], the headroom of GLUE is drastically decreasing. SuperGLUE [61] was later proposed as a more rigorous test for language understanding, which incorporates more challenging and diverse tasks. XTREME [23] and XGLUE [37] have also been proposed for benchmarking multilingual language understanding. Our VALUE benchmark shares similar merit to these language understanding benchmarks, focusing on understanding and generation tasks in the video-and-language domain.

Compared to the blossoming of natural language benchmarks, video-and-language (VidL) understanding still lacks a large-scale benchmark to systematically track advances in this area. Methods

---

[3]ASR can be applied when subtitles are not available.

**Table 2: Statistics of datasets** in VALUE benchmark. Ground-truth annotations on Test (leaderboard) split are hidden from the public, and used to rank model performance. (†) VIOLIN and VLEP are 2-way classification tasks, which are considered as special QA tasks in our benchmark for simplicity. *AveR* denotes Average Recall at {1, 5, 10}, *Acc.* = Classification Accuracy.

| Task | Dataset | Data Statistics (# videos/ # queries, QAs, captions) | | | | Metrics |
|---|---|---|---|---|---|---|
| | | Train | Val | Test (public) | Test (leaderboard)[4] | |
| Retrieval | TVR [35] | 17.4K/87.1K | 2.2K/10.9K | - | 2.2K/10.9K | AveR |
| | How2R [36] | 21.3K/27.1K | 1.0K/1.3K | - | 1.0K/1.3K | |
| | YC2R [74] | 10.3K/10.3K | 3.5K/3.5K | - | 1.6K/1.6K | |
| | VATEX-EN-R [63] | 26.0K/259.9K | 3.0K/30.0K | - | 6.0K/60.0K | |
| QA | TVQA [32] | 17.4K/122.0K | 2.2K/15.3K | - | 2.2K/15.3K | Acc. |
| | How2QA [36] | 24.5K/34.2K | 3.1K/3.1K | - | 3.1K/3.1K | |
| | VIOLIN† [40] | 12.7K/76.1K | 1.6K/9.6K | 1.6K/9.6K | 1.3K/7.7K | |
| | VLEP† [34] | 7.2K/20.1K | 1.6K/4.4K | | 1.5K/4.2K | |
| Captioning | TVC [35] | 17.4K/86.7K | 10.8K/43.6K | - | 10.8K/43.6K | CIDEr-D |
| | YC2C [74] | 10.3K/10.3K | 3.5K/3.5K | - | 1.6K/1.6K | |
| | VATEX-EN-C [63] | 26.0K/259.9K | 3.0K/30.0K | 6.0K/60.0K | 6.2K/62.8K | |

developed [57, 76, 45, 41, 26, 29, 36, 20, 50, 31, 58, 44, 39, 73] in this field are often evaluated on different tasks, datasets and experimental settings, making fair comparison difficult. The Pentathlon Challenge [6] held at CVPR 2020 combines 5 text-based video retrieval tasks to compare models using a set of pre-extracted expert features. However, the Challenge only focuses on a single task, and limits the models to only using offline extracted features. In contrast, VALUE is designed to incorporate a diverse set of tasks, including text-based video retrieval [74, 63], video moment retrieval [35, 36], video question answering [32, 36], video captioning [74, 63, 35], video-and-language inference [40], and next event prediction [34]. Meanwhile, VALUE is model-agnostic and welcomes methods of all kinds.

## 3 VALUE Benchmark Tasks

VALUE aims to provide a one-stop evaluation for multi-channel video understanding on 3 common video-and-language (VidL) tasks: ($i$) text-based video retrieval; ($ii$) video question answering (QA); and ($iii$) video captioning. To construct a comprehensive evaluation benchmark, we include recent datasets collected on multi-channel videos: TVR [35], How2R [36], TVQA [32], How2QA [36], VIOLIN [40], VLEP [34] and TVC [35]. Since most of these datasets focus on understanding long videos in TV/movie domain, we further select another two popular datasets, YouCook2 [74] and VATEX [63], originally built on shorter single-channel YouTube videos, to cover diverse video genres and lengths. In total, VALUE assembles 11 diverse VidL datasets.[5] There are other VidL datasets on single-channel videos that are not included in VALUE, due to the difficulties in collecting hidden test set [11, 70, 8], unnatural annotations [69], or the lack of subtitle channel in GIF videos [25].

Table 1 summarizes the statistics of video data provided in VALUE. The videos come from diverse domains, ranging from TV episodes and movie clips with different temporal dynamics, event shifts and people interactions, to instructional videos and vlogs dominated by monologues with less human-centered scenes. Average video length varies from 10 to 76 seconds. All the video datasets except VATEX [63] have a high multi-channel ratio (proportion of videos to subtitles). Table 2 summarizes the selected tasks and datasets. Figure 1 shows an illustration of the VALUE benchmark.

Our VALUE evaluation server is hosted on CodaLab.[6] In the following subsections, we will introduce each task. The benchmark site shows the scores per-task and a meta-average of those scores across all tasks to determine a system's rank on the leaderboard.

---

[4]The ground-truth annotations on Test (leaderboard) split are either obtained from the author or collected following the same procedure as in the original paper.

[5]Video features, subtitles and annotations for all the VALUE tasks are released at https://github.com/value-benchmark/DataRelease. Due to copyright issue, we are unable to publish raw videos. However, we provide all the YouTube ids/TV episode versions along with their original timestamps to facilitate end-to-end training on VALUE benchmark.

[6]See our submission page for details: https://value-benchmark.github.io/submission.html.

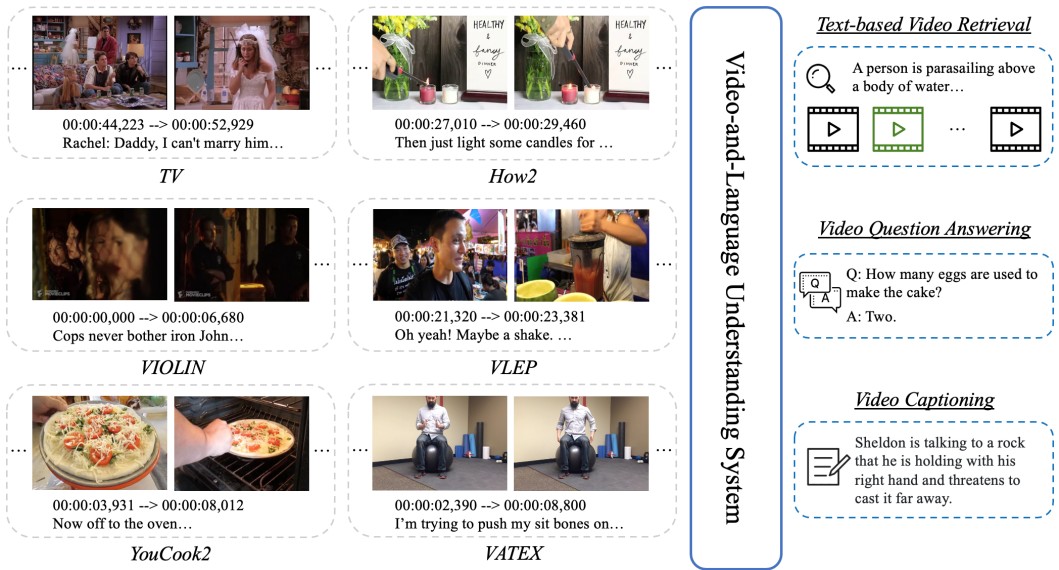

**Figure 1:** An illustration of VALUE benchmark. VALUE provides a one-stop evaluation for multi-channel video understanding on 3 common video-and-language tasks: ($i$) text-based video retrieval; ($ii$) video question answering; and ($iii$) video captioning. The videos in VALUE come from 6 data sources, covering diverse genres and domains.

## 3.1 Text-based Video Retrieval Tasks

In VALUE, there are two types of text-based video retrieval tasks: ($i$) Video Corpus Moment Retrieval (VCMR): TVR and How2R datasets; and ($ii$) Video Retrieval (VR): YouCook2 Retrieval (YC2R) and VATEX Retrieval (VATEX-EN-R) datasets. VR requires a model to retrieve the most relevant video clip from the video corpus described by the textual query. VCMR is more challenging, requiring a model to not only retrieve the most relevant video clip from the video corpus, but also locate the relevant moment in the retrieved video clip. Stand-alone evaluation on temporal moment localization tasks [7, 21] is not included in our benchmark, as the two VCMR tasks already evaluate the ability of the model to localize relevant moment as a sub-task. The upper block of Table 2 summarizes the statistics of the 4 datasets for retrieval tasks.

**TVR** [35] consists of 109K queries on 21.8K videos from 6 TV shows of diverse genres, where each query is associated with a tight temporal alignment. Among all queries, 74.2% are related to video only, 9.1% to text only, and 16.6% to both video and text. The dataset is divided into 80% train, 10% val, 5% test-public, and 5% test-private. We combine the test-public set with the test-private set for leaderboard evaluation.

**How2R** [36] is collected following the same procedure of TVR, but based on 60-second clips from 9K instructional videos in HowTo100M [46], on average 2-3 queries per clip. The original How2R data are noisy due to short and repetitive textual queries. For VALUE benchmark, we remove queries with fewer than 6 words and repetitions. After cleaning, 2K video clips and the associated queries are held-out for validation and testing, and the rest for training.

**YouCook2 Retrieval (YC2R)** [74] consists of 2K YouTube cooking videos across 89 recipe types. The videos are split into a 67%/23%/10% for training/validation/testing and segmented by human annotators into clips that represent recipe steps. Each clip is annotated with one textual description. We follow [46] to evaluate retrieval performance at clip level. We obtain private labels from the authors to provide evaluation on test set, and augment the videos with ASR-generated subtitles.

**VATEX Retrieval (VATEX-EN-R)** VATEX [63] was originally developed for multilingual video captioning and video-guided machine translation tasks. It contains 41.3K videos of 600 fine-grained human activities and 825K captions in both English and Chinese. To ensure its consistency with other tasks being considered, we take videos and English captions to evaluate retrieval performance. Videos are split into 26K/3K/6K/6K for training/validation/public testing/private testing. We use the

public test set to evaluate retrieval performance, to avoid potential data leaks on private testing set of the original VATEX tasks. We also provide ASR-generated subtitles.

To evaluate the model performance, we adopt the average recall at K (R@K) over all queries as the metric. For VR (*i.e.*, YC2R and VATEX-EN-R), we consider a prediction correct if the predicted video matches the ground-truth video. For VCMR (*i.e.*, TVR and How2R), we additionally require that the predicted span of a correct prediction has a high overlap with the ground-truth moment. We use temporal Intersection over Union (tIoU) to measure the overlap between the predicted span and the ground-truth span.[7] We use AveR (the average of R@{1, 5, 10}) as the final metric to evaluate model performance on retrieval tasks.

## 3.2 Video Question Answering Tasks

We group tasks that use classification accuracy as the evaluation metric into video question answering (QA) tasks. The middle block of Table 2 summarizes the data statistics of 4 datasets considered.

**TVQA** [32] is collected under multiple-choice settings from TV videos. Each video clip contains 7 questions, with 5 answers per question. The start/end points of the relevant moments are also provided for each question. TVQA consists of 3 sub-tasks: (*i*) QA on the grounded clip; (*ii*) question-guided moment localization; and (*iii*) QA on the full video clip. We only consider QA on the full video clip, as this is the most challenging setting among the three. We combine the test-public set with the test-private set for leaderboard evaluation.[8]

**How2QA** [36] was collected in a similar way to TVQA, but on video clips sampled from instructional videos in HowTo100M [46]. Each video clip is annotated with an average of 1-2 questions, with 4 answers per question. Similarly, the questions in How2QA are grounded temporally, but we only consider QA on the full video clip. As the video clips used in How2QA largely overlap with those in How2R, we re-split the video clips and their associated QA pairs into 80% train, 10% val and 10% test, to avoid potential data leaks.

**VIOLIN** [40] is introduced as a new video-and-language inference task. Given a premise video clip with aligned subtitles and a hypothesis sentence, the task is to predict whether the premise entails the hypothesis or contradicts the hypothesis. Its original release consists of 95.3K video-hypothesis pairs with ground-truth annotations from 15.9K video clips, split into 80% train, 10% val and 10% test. We further collect a hidden test split (*i.e.*, Test (leaderboard) in Table 2) with 4K hypothesis on 1.5K video clips from the same video domain for leaderboard evaluation.

**VLEP** [34] is a dataset for video-and-language commonsense-based future event prediction. Given a video with aligned subtitles, the task is to choose which of the two future events is more likely to occur after that video. VLEP contains 28.7K future event prediction examples from 10.2K TV shows and YouTube Lifestyle Vlog video clips, which are split into 70% train, 15% val and 15% test.

## 3.3 Video Captioning Tasks

For video captioning tasks, we consider 3 datasets (lower block of Table 2).

**TVC** [35] is a multi-channel video captioning dataset extended from TVR, containing 262K descriptions paired with 108K video moments. Unlike traditional video captioning tasks, the descriptions are collected on video moments instead of the entire video, and video subtitles are used as additional model input. For a given video and the start/end points of a moment of the video, a model must generate a description for the video moment with/without leveraging the information from the entire video. We combine the test-public set with the test-private set for leaderboard evaluation.

**YouCook2 Captioning (YC2C)** [74] is built on the same cooking videos as in YouCook2 Retrieval task. Each video clip is annotated with one captioning sentence. Depending on whether we regard each clip individually or combine the clip captions into a paragraph, the evaluation for each video can be either at clip-level, by reporting the averaged score on each clip over the entire video corpus; or at video-level, evaluating each merged paragraph. We follow [75] to evaluate clip-level performance, and to maintain consistency with other captioning datasets considered. The test set is used for leaderboard evaluation.

---

[7]During evaluation, the average recalls are measured by tIoU$\geq$0.7.

[8]Train, val and test video splits are the same as TVR.

**VATEX Captioning (VATEX-EN-C)** [63] Similar to VATEX Retrieval, we take videos and English captions in VATEX as another task to evaluate video captioning on multi-channel videos. Each video is annotated with 10 English captions, with 5 regular English captions and 5 parallel English captions translated from Chinese. The private test set is used for leaderboard evaluation.

The performance of video captioning tasks is measured by comparing predicted captions against corresponding ground-truth captions, with standard captioning metrics applied (*e.g.*, BLEU@4 [49], METEOR [15], ROUGE-L [38], and CIDEr-D [60]). We use the CIDEr-D score as the main metric to evaluate model performance.

# 4 Experiments and Analysis

In this section, we provide extensive experiments and analysis to demonstrate the value of VALUE benchmark. Specifically, we investigate the impact of input channels and video-subtitle fusion methods (Sec. 4.2), evaluate the effectiveness of different visual representations (Sec. 4.3), and study the transferability between tasks (Sec. 4.4) and the impact of multi-task learning (Sec. 4.5).

## 4.1 Features and Baseline

**Multi-Channel Video Representations.** The context inputs are multi-channel videos, *i.e.*, videos and their associated subtitles. For *subtitle channel*, we follow [42] and tokenize each subtitle sentence into a sequence of WordPieces [66]. The final representation for each sub-word token is the summation of its token embedding and position embedding, followed by a layer normalization (LN) layer. For *video channel*, we extract 2D appearance features and 3D motion features every 1.5 seconds. We use ResNet-152 [22] pre-trained on ImageNet [14] to extract 2D features, and use SlowFast [19] pre-trained on Kinetics [27] to extract 3D features. These features are concatenated and fed through a fully-connected (FC) layer to be projected into the same lower-dimensional space as token embeddings. Since video frames are sequential, their position embeddings can be calculated in the same way as word tokens. The final embedding of a video segment is obtained by summing up FC outputs and position embeddings, then passing through an LN layer.

**Baseline Architecture.** There are many pioneering works on building generalizable VidL understanding systems via large-scale pre-training [46, 76, 45, 57, 36]. However, most focus on single-channel videos, thereby cannot be evaluated directly on or easily extended to multi-channel videos. Our selected baseline architecture is based on HERO [36], due to its strong capacity of understanding multi-channel videos and its generalizability to different VidL tasks.[9]

HERO takes as inputs a sequence of video segments and subtitle sentences, and encodes them in a hierarchical fashion, with a cross-modal transformer to fuse subtitle sentences and their accompanying local video segments. The cross-modal transformer is followed by a temporal transformer to obtain a globally contextualized embedding for each segment, using all the segments in the video. HERO can be applied to different types of VidL tasks as a multi-channel video encoder. To evaluate on VALUE tasks, we perform task-specific adaptation by adding different task heads. See Appendix for more details.

**Pre-training.** We directly adopt the pre-trained checkpoint released in HERO, which was pre-trained on over 7M video clips from HowTo100M [46] and TV dataset [32], with 4 pre-training tasks, *e.g.*, Masked Language Modeling and Video-Subtitle Matching. For finetune-only experiments, model parameters are initialized with pre-trained RoBERTa weights [42, 65].

## 4.2 Impact of Input Channels and Video-Subtitle Fusion Methods

In this section, we investigate how information from both video and subtitle channels can be used effectively in multi-channel videos. Specifically, we try to answer the following questions:

**Q1: Is video or subtitle channel alone sufficient to achieve good performance?** Most previous works only leverage visual information from the video channel [46, 76]. To assess the importance of the subtitle channel, we evaluate and compare three models: ($i$) video-only, where the model takes only visual features as input; ($ii$) sub-only, where the model takes only subtitle sentences as input; and ($iii$) video+sub, where the model takes both visual features and subtitle sentences as input.

---

[9]Code is released at https://github.com/value-benchmark/StarterCode.

**Table 3:** Impact of **input channels**. For video-only experiments, we replace all subtitle texts with empty strings. For sub-only experiments, the visual features are replaced with zero vectors. All results are reported on Val/Test (public) split without pre-training.

| Input Channel | TVR | How2R | YC2R | VATEX-EN-R | TVQA | How2-QA | VIO-LIN | VLEP | TVC | YC2C | VATEX-EN-C | Meta-Ave |
|---|---|---|---|---|---|---|---|---|---|---|---|---|
| | AveR | AveR | AveR | AveR | Acc. | Acc. | Acc. | Acc. | C | C | C | |
| Video-only | 4.49 | 1.70 | 9.74 | 57.50 | 44.17 | 60.42 | 58.53 | 57.56 | 37.52 | 53.61 | 51.14 | 39.67 |
| Sub-only | 1.95 | 0.98 | 32.31 | 5.21 | 70.15 | 68.15 | 66.26 | 58.06 | 38.74 | 93.33 | 9.28 | 40.40 |
| Video+Sub | **7.72** | **1.91** | **33.91** | **58.99** | **71.08** | **69.44** | **66.83** | **58.79** | **48.48** | **108.46** | **52.15** | **52.52** |

**Table 4:** Impact of **video-subtitle fusion methods**. Refer to Section 4.2 for detailed explanation of each method. HERO's fusion method can also be expressed as *temp. (temporal) align + cross-modal transformer*. All results are reported on Val/Test (public) split without pre-training.

| Fusion Method | TVR | How2R | YC2R | VATEX-EN-R | TVQA | How2-QA | VIO-LIN | VLEP | TVC | YC2C | VATEX-EN-C | Meta-Ave |
|---|---|---|---|---|---|---|---|---|---|---|---|---|
| | AveR | AveR | AveR | AveR | Acc. | Acc. | Acc. | Acc. | C | C | C | |
| 1 two-stream | 5.66 | 1.90 | 32.60 | 48.19 | **71.15** | **69.63** | 66.61 | 58.49 | 42.67 | 99.35 | 39.04 | 48.66 |
| 2 sequence concat | 5.60 | 2.73 | **35.55** | **60.24** | 69.61 | 68.99 | 66.09 | **60.91** | 44.73 | 99.78 | **52.65** | 51.53 |
| 3 temp. align + sum | 6.75 | 2.44 | 31.84 | 58.11 | 70.23 | 69.44 | 66.33 | 57.72 | 47.80 | 104.97 | 52.07 | 51.61 |
| 4 temp. align + concat | 7.10 | **3.19** | 32.59 | 57.33 | 69.81 | 69.31 | 66.16 | 58.54 | 47.12 | 100.90 | 52.09 | 51.29 |
| 5 HERO | **7.72** | 1.91 | 33.91 | 58.99 | 71.08 | 69.44 | **66.83** | 58.79 | **48.48** | **108.46** | 52.15 | **52.52** |

Results are summarized in Table 3. When leveraging both video and subtitle channels, the model achieves the highest meta-average score (52.52) with the best performance across all VALUE tasks.

We also observe that QA tasks generally benefit more from subtitle channel than video channel, but not so for retrieval and captioning tasks. For tasks collected on multi-channel videos (*i.e.*, TV and How2 videos), the model needs to exploit information from both channels to achieve the best performance. For tasks that are originally collected without subtitle channel (*i.e.*, YC2 and VATEX videos), adding subtitle channel still helps. Especially for YC2 tasks (YC2R and YC2C), the subtitle-only model performs significantly better than video-only model. This is not surprising, as the cooking steps are often clearly described in the dialogues/monologues of cooking videos, so that there is a higher correlation between the retrieval query and the caption. Vice versa, VATEX tasks rely more on video channel than subtitle channel, as the 10-second videos in VATEX focus more on human activities and half of them have no subtitles.

**Q2: How to effectively fuse video and subtitle embeddings?** To answer this, we propose several model variants based on HERO, and compare their performance in Table 4. The simplest baseline is a two-stream architecture [55, 32], where the video segments and subtitle sentences are processed separately with different streams to obtain a modality-specific prediction. The final prediction is the average of the predictions from the two streams. Such a late fusion method results in the worst performance (meta-average score of 48.66), as the predictions based on the single-channel inputs are independently modeled without considering information from the other channel.

We further investigate 3 simple ways to fuse video and subtitle embeddings at an earlier stage. The three baseline methods are: (*i*) *sequence concat*, concatenating embeddings at sequence level without temporal alignment; (*ii*) *temp. (temporal) align + sum*, summation of the temporally aligned video segment embeddings and subtitle sentence embeddings; and (*iii*) *temp. align + concat*, concatenation of the temporally aligned video segment embeddings with subtitle sentence embeddings at feature level. Finally, we compare with the video-subtitle fusion method proposed in HERO, where the temporally aligned video segments and subtitle sentence tokens are fed into the cross-modal transformer to compute the contextualized embeddings for each video segment. These fused embeddings from all the methods above are then fed into the same temporal transformer to learn the global video context and obtain the final video embeddings.

As shown in Table 4, HERO achieves the highest meta-average score (52.52), but its performance is sub-optimal on some tasks. For example, the best performance on VATEX tasks is achieved by *sequence concat*, which also outperforms HERO on How2R, YC2R and VLEP. We speculate that the joint video and subtitle representations for those relatively short videos (*e.g.*, VLEP and VATEX) can also be modeled well by simply concatenating VidL embeddings at sequence level, without explicitly

Table 5: **Task transferability.** We train model on one task and test it on another task of the same task type. All results are reported on Val/Test (public) split without pre-training. The best and second best performance are highlighted with bold and underline, respectively.

**(a)** Retrieval Tasks.

| Train Data | TVR | How2R | YC2R | VATEX-R |
|---|---|---|---|---|
| TVR | **7.72** | 0.00 | 0.35 | 2.79 |
| How2R | 0.03 | **1.91** | 10.30 | 10.31 |
| YC2R | - | - | **33.91** | 1.01 |
| VATEX-R | - | - | 3.82 | **58.99** |

**(b)** QA Tasks.

| Train Data | TVQA | How2QA | VIOLIN | VLEP |
|---|---|---|---|---|
| TVQA | **71.08** | 36.89 | 50.01 | 53.23 |
| How2QA | 21.75 | **69.44** | 53.85 | 55.65 |
| VIOLIN | 20.12 | 40.55 | **66.83** | 44.26 |
| VLEP | 22.16 | 26.04 | 50.00 | **58.79** |

**(c)** Captioning Tasks.

| Train Data | TVC | YC2C | VATEX-C |
|---|---|---|---|
| TVC | **48.48** | 1.35 | 1.72 |
| YC2C | 0.43 | **108.46** | 0.74 |
| VATEX-C | 4.25 | 7.09 | **52.15** |

aligning them on the temporal domain. Note that the goal is to find a generalizable video-subtitle fusion method that can perform well across 11 VALUE tasks. Therefore, we use HERO as the optimal method for future experiments.

## 4.3 Impact of Visual Representations

The common practice to represent a video [35, 57] is to extract 2D appearance features from pre-trained 2D models (*e.g.*, ResNet [22]) and 3D motion features from 3D models (*e.g.*, SlowFast [19]) at the same fixed frame rate, then concatenate them together. In this section, we investigate the impact of using different visual representations for videos.[10]

We leverage several pre-trained models to extract video features. For 2D appearance features, we start with the widely adopted ResNet(-152) [22] pre-trained on ImageNet [13]. Recent work [31, 44, 9] show that with image-text pre-training, models trained on 2D features alone can achieve decent performance on many video-and-language (VidL) tasks. Thus, we further evaluate 2D features generated by ViT [17] in CLIP [51], which is pre-trained with a large-scale image-text corpus. For 3D motion features, we also evaluate two variants, one from Kinetics [27] pre-trained SlowFast [19] model and the other from an S3D model [67] pre-trained on 100M video-text pairs [45]. In addition, we explore different combinations of these features by concatenating 2D features with 3D features following common practice. Through this investigation, we hope to understand whether VALUE tasks are designed to favor 2D appearance information from sparsely sampled frames, or require 3D motion information from dense video frames, or rely on both to accomplish these tasks. Results are presented in Appendix. Without pre-training, we found that the best performance is achieved by CLIP-ViT+SlowFast, suggesting that both appearance and motion information are required to handle VALUE tasks. With pre-training (from HERO's pre-trained checkpoint, trained using ResNet+SlowFast features), ResNet+SlowFast achieves the best performance, likely due to the better matched pre-training and finetuning setting.[11] In the following, we base all of our experiments on ResNet+SlowFast.

## 4.4 Task Transferability Evaluation

In this section, we study how VALUE tasks relate to each other. Specifically, we train model on one task and test it on another task of the same type. Results are summarized in Table 5. Across all task types, the absolutely low performance when transferring the model trained on one task to another indicates that there are significant differences between tasks. The differences can be caused by domain gaps (*e.g.*, TV videos in TVQA and instructional videos in How2QA), discrepancies in video length (*e.g.*, model trained on 60-90 seconds long videos in TVC may not work well on 10-second long videos in VATEX-EN-C) and different task formalization (*e.g.*, model trained on YC2R cannot directly apply to TVR). These results in turn suggest that VALUE supports diverse VidL tasks, thus providing a comprehensive evaluation for VidL understanding systems.

## 4.5 Multi-Task Learning Evaluation

The low performance observed in the transfer evaluation of task-specifically trained models leads to a natural question: *can one model conquer them all?* In this section, we investigate several multi-task learning baselines and report the results in Table 6. We first establish the baseline performance by

---

[10]All visual features are released to reproduce the experimental results in this section.

[11]We did not perform pre-training using other visual features due to its enormous computation cost.

**Table 6:** Evaluation of **multi-task learning baselines** on Test (leaderboard) set. Results are reported on HERO architecture with ResNet+SlowFast features. We compare the following model training settings: single-task training (ST), multi-task training (MT) by tasks or domains, all-task training (AT) and AT first then ST (AT → ST). The best performance (of each block) are highlighted with bold (underline).

| Training Setting | TVR | How2R | YC2R | VATEX-EN-R | TVQA | How2-QA | VIO-LIN | VLEP | TVC | YC2C | VATEX-EN-C | Meta-Ave |
|---|---|---|---|---|---|---|---|---|---|---|---|---|
| | AveR | AveR | AveR | AveR | Acc. | Acc. | Acc. | Acc. | C | C | C | |
| 1  Human | - | - | - | - | 89.41 | 90.32 | 91.39 | 90.50 | 62.89 | - | 62.66 | - |
| *Finetune-only* | | | | | | | | | | | | |
| 2  ST | 7.70 | 1.74 | 40.69 | 38.34 | 70.54 | 69.00 | 63.75 | 57.94 | 46.76 | 106.24 | 52.16 | 50.44 |
| 3  MT by Task | 7.75 | 1.90 | 46.38 | 38.17 | 71.26 | 71.43 | 64.74 | 68.01 | 46.01 | 105.22 | 51.07 | 52.00 |
| 4  MT by Domain | 10.01 | 2.69 | 44.58 | 36.10 | 73.94 | 70.01 | 65.93 | 67.37 | 46.53 | 100.74 | 50.46 | 51.97 |
| 5  AT | 9.76 | 2.42 | 47.91 | 37.33 | 73.98 | 71.14 | 65.80 | 68.03 | 46.46 | 101.72 | 51.07 | 52.33 |
| 6  AT→ST | 10.43 | 2.68 | 49.48 | 38.58 | 73.46 | 71.88 | 65.73 | 67.80 | 46.12 | 103.73 | 51.87 | 52.89 |
| *Pre-train + Finetune* | | | | | | | | | | | | |
| 7  ST | 12.04 | 4.09 | 57.88 | 40.63 | 74.36 | 74.76 | 65.31 | 68.46 | 48.97 | 127.94 | 52.57 | 57.00 |
| 8  MT by Task | 12.63 | 4.66 | 59.20 | 39.97 | 74.56 | 74.40 | 66.34 | 68.11 | 48.02 | 123.40 | 50.49 | 56.53 |
| 9  MT by Domain | 11.53 | 4.03 | 52.14 | 36.97 | 74.54 | 74.08 | 65.92 | 68.06 | 47.23 | 100.29 | 45.95 | 52.79 |
| 10  AT | 11.61 | 4.03 | 52.20 | 38.01 | 75.12 | 73.66 | 66.60 | 68.27 | 46.04 | 109.11 | 49.74 | 54.04 |
| 11  AT→ST | 12.17 | 4.51 | 54.16 | 38.86 | 75.05 | 74.24 | 66.93 | 67.96 | 46.38 | 120.86 | 50.59 | 55.61 |

training single-task models on HERO architecture for each of the 11 datasets (ST, L2). We also include human performance (L1) on eligible QA and captioning tasks.[12] Next, we compare different multi-task learning baselines.

**Multi-Task Learning by Task Type.** We begin our investigation with the most intuitive setting - jointly training tasks within the same task type (MT by Task, L3). As the tasks of the same type are typically highly related, this is akin to some data augmentation practice. Note that this corresponds to 3 separate multi-task models - one for each task type. Comparing to ST models (L2), we see that MT by Task achieves +1.56 points improvement on meta-average score (52.00 *vs.* 50.44). The increase in meta-average score results from performance improvements on retrieval and QA tasks, with larger improvements on tasks with smaller-scale data (*e.g.*, YC2R and VLEP).

On captioning tasks, multi-task learning results in a slight performance degradation. Note that a single decoder is shared among the three captioning tasks, with task-specific vocabularies combined together. This combined vocabulary may introduce more noise than single-task learning when applied to a specific captioning dataset. Similar performance decrease is consistently observed in other multi-task learning baselines. For simplicity, we leave out discussions on captioning results.

**Multi-Task Learning by Domain.** We explore another multi-task learning setting, where we jointly train tasks within the same domain (MT by Domain, L4). We first divide the 11 datasets into 2 domains based on video genre: (*i*) TVR, TVQA, VIOLIN, VLEP and TVC are grouped into TV domain, and (*ii*) the rest of the datasets are grouped into YouTube domain. Note that the videos in TV domain largely overlap among different datasets. However, for datasets in YouTube domain, their videos cover more diverse contents (*e.g.*, YC2 videos focus on cooking while VATEX videos present a wide range of human activities). Compared with ST (L2), MT by Domain improves by +1.53 on meta-average score (51.97 *vs.* 50.44). In TV domain, model performance improves significantly, suggesting that these different tasks require similar understanding about TV videos. Under YouTube domain, we observe similar improvements on most of the datasets except VATEX-EN-R, where the model seems to be over-fitting to the validation split (Table **??** in Appendix).

**All-Task Learning.** We switch to the "extreme" multi-task setting - a single model trained on all 11 datasets (AT, L5). This model outperforms separtely trained ST models (L2) for 8 out of 11 tasks and improve the meta-average score by +1.89 points (52.33 *vs.* 50.44), while the number of parameters are significantly reduced by approximately 11 times. Our AT model also outperforms the other two multi-task baselines (L3-4) on meta-average score despite having fewer parameters. This implies that, despite their diversity, tasks across different task types and domains can benefit from joint training.

**Multi-Task Learning as Pre-training.** Finetuning from a multi-task trained model allows the model to take advantage of the additional, diverse supervision captured during multi-task training.

---

[12]See Appendix for more information on human evaluation.

Following [43], we explore finetuning each task (AT → ST, L6) from the multi-task learned weights (L5). Results show that this strategy further improves meta-average by +0.56 points (52.89 *vs.* 52.33).

**Combining Multi-Task Finetuning with Pre-training.** In L7-11, we take advantage of pre-trained HERO model and repeat experiments in L2-6. When compared with their counterparts without pre-training, we observe consistent performance improvements across all training settings considered. However, pre-training and multi-task finetuning often do not complement each other. Especially on captioning tasks, the performance degradation from multi-task finetuning is even more severe. The best performance is achieved in single-task finetuning with a meta-average score of 57.00 (L7). However, our best model is still far from achieving human parity (L1), especially on QA tasks.

## 5  Conclusion and Discussion

We introduce VALUE, a comprehensive benchmark for evaluating video-and-language (VidL) understanding systems. VALUE includes 11 VidL datasets with multi-channel video inputs over 3 popular tasks, covering a wide range of video genres, video lengths, task difficulties and data volumes. Through extensive experiments, we conclude that designing general-purpose VidL models still remains challenging. We believe that VALUE provides fertile soil for addressing this challenge. For future work, we plan to add diagnostic datasets and support analysis of submitted models both quantitatively and qualitatively, to provide more insights into pushing the state of the art on VALUE.

Although we aim for a comprehensive video-and-language evaluation benchmark, as discussed in Section 3, our benchmark currently only contains a selected set of datasets and tasks. It is worthwhile to add more eligible datasets and tasks (considering their diversity, difficulty, etc.) as the next step of the benchmark. Meanwhile, due to the limited availability of multilingual VidL datasets, all datasets covered in the benchmark are of a single language (*i.e.*, English). Future work could consider multilingual VidL datasets [63, 54, 24, 30] as a complementary evaluation to further test systems' ability on processing information in different languages.

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
