# OpenReview forum: "VALUE: A Multi-Task Benchmark for Video-and-Language Understanding Evaluation"
_NeurIPS.cc/2021/Track/Datasets_and_Benchmarks/Round1 — NeurIPS 2021 Datasets and Benchmarks Track (Round 1)_

### Official Review · Reviewer_sp5p · 2021-07-04

**Rating:** 7
**Confidence:** 4
**Clarity:** The paper is well written and explain…

**Strengths:**

1) This is the first work to propose a benchmark for evaluating video-and-language models on different tasks. As research on multi-task learning and video-language pretraining is making great progress recently, this new benchmark is of high important to the research field.
2) The datasets are of high diversity, and current methods can not perform very well on the evaluation metrics, indicating that there is much work to be done and more advanced approaches are necessary.
3) There are many experiments and analyses on components of the baseline models, as well as the transferability and multi-task learning. Those experiments provide inspiring information and will shed light on future research.
4) Unified evaluation metrics, pre-extracted features, benchmark, starter code, and leaderboard are provided, so the benchmark is well ready to be investigated and make an impact to researches in this field.

**Weaknesses:**

1) It would be good to provide some statistics or intuitive explanations on the diversity of those videos. For example, the distribution of the video lengths and keywords.
2) What is the meta-average score? Is it the average of evaluation scores on all tasks and datasets? If so, I think there are some potential drawbacks of this metric. Evaluation scores for different tasks may have different ranges. When averaging the scores of different tasks, the averaged score may be dominant by the one with the largest range. For example, in Table 2 the Video QA accuracy has a small range compared to retreival and captioning scores, so the difference in VQA performance might be neglected in the final meta-average score. Since the scores of different evaluation metrics are not comparable in numbers, I doubt does it make sense to take the average of them?
3) When averaging the scores of different datasets to calculate the meta-average score, is the size of test sets taken into consideration? For example, the YC2C leaderboard test set contains only 1.6K videos, while the TVC leaderboard test set contains 10.8K videos. When calculating the meta-average, are the scores of those datasets simply averaged or weighted averaged?
4) The retrieval performance AveR of How2R dataset is very low. Does it mean that the retrieval task defined on this task is impossible? Are there issues with the task definition on this dataset?

**Additional Feedback:**

I am impressed by the new benchmark, and I believe that it will make a high impact on video-and-language research. My major concern is the definition of meta-score seems unreasonable and may lead to undesired behaviors. The meta-score may be dominated by tasks whose evaluation metrics have the highest range. Also, samples from small datasets will get higher weights when calculating the meta-average score. If the authors can address or explain this in the rebuttal, I will be happy to raise the score.

**Post-rebuttal Updates**
The authors addressed my concerns well. I am convinced by the discussions on the meta-score and the proposed average rank score in the rebuttal. So I raised my rating to acceptance.

**Correctness:**

I have doubt about the meta-average score which takes the average number of different evaluation metrics. It seems not very reasonable to take average of different metrics since numbers of different evaluation metric scores are not comparable. Also, the meta-averaged score may be dominated by tasks whose evaluation scores have large ranges and neglect tasks whose evaluation scores have small ranges. In addition, by taking the average of different datasets, the samples in those datasets with a smaller test set will get higher weight on the meta-average score.

**Documentation:**

There is sufficient detail.

**Ethics:**

There are no ethical concerns.

**Relation To Prior Work:**

It is clearly discussed how this work differs from previous contributions.

**Summary And Contributions:**

This paper introduces a new benchmark for video-and-language understanding evaluation. It assembles 11 video-and-language datasets over three tasks, text-to-video retrieval, video question answering, and video captioning. The benchmark contains a diverse collection of videos with their subtitles. Simple evaluation metrics are also defined for the three tasks. Various baseline methods are evaluated, including models with and without pretraining, different video input channels, different video representations, and fusion methods. Task transferability evaluation and multi-task learning evaluation are also conducted. This paper is the first work to provide a unified benchmark and evaluation for different video-and-language tasks. It also provides analysis on different baseline methods. I think it will shed light on future research on video-and-language, and will make significant contributions to this field.

---

> ### Author Response · Authors · 2021-07-14
> **Thanks for the insightful feedback, we address each of the comments below.**
>
> `The distribution of the video lengths and keywords`
>
> Thanks for your suggestion. We have added the distributions of video lengths and top-20 nouns and verbs for each dataset in the updated Appendix Section A.
>
>
> `Meta-Average Score`
>
> Thanks for your insightful comments.
>
> Meta-average (macro-average) score is an unweighted average of evaluation scores across all tasks and datasets. We agree that the meta-average score is not a perfect metric, due to different scales of evaluation metrics (e.g., CIDER for captioning and accuracy for QA) and different data scales (e.g., differences in video corpus size for YC2R and VATEX-EN-R). However, the weighted average score based on data scales (i.e., micro-average) may be dominated by tasks/datasets that have larger scales (e.g., VATEX), and also does not consider different scales of evaluation metrics.
>
> For easier comparison between baseline models in our paper, we follow GLUE and SuperGLUE to use macro average (namely, an unweighted average of the scores on each dataset) instead of micro average (weighted average score based on data scales).  To ensure a fair comparison between models, VALUE leaderboard provides evaluation of models at different levels: each dataset, each task, all datasets+all tasks. We encourage future work on the VALUE benchmark to report results at different levels, to provide a comprehensive evaluation of model performance.
>
> To address concerns about meta-average brought up by the reviewer, we had a fierce discussion among the authors. We are planning to use the average rank of model ranks on each dataset to decide the overall rank of models on our leaderboard while keeping the meta-average score as reference. To some extent, it is similar to ranking different countries/regions among different competitions for the Olympics game. Average rank score should promote models that are capable of performing well on each dataset/task in a more balanced way, eliminating the bias towards larger/smaller scale datasets or the range of evaluation metrics. In the meantime, we will also look into adding more metrics, especially via feedback from the community (e.g., range-normalized meta-average).
>
> `How2R Performance`
>
> How2R was collected following the collection process of TVR, but on different video domains, i.e., How2 videos. The task is defined the same as TVR. Thus, we do not expect any issue with the dataset and task definition. We conjecture the low performance on How2R may be due to the fine-grained scene change in instructional videos. For example, for a single cooking video, “mixing eggs in a bowl” and “mixing flour with the egg” only differs in the content of the bowl while the majority of the scene stays the same, this makes moment retrieval in a single video very hard. However, current video feature extraction is based on image level/video segment level instead of object level. Therefore, it is hard to capture such differences.

---

### Official Review · Reviewer_EXqq · 2021-07-05
**A comprehensive dataset  is built through unifying existing datasets but lacks novelty and more detailed processing.**

**Rating:** 5
**Confidence:** 3

**Strengths:**

1.	This dataset could serve for pre-training large deep models.
2.	This dataset is able to support various datasets, making it possible to test a method from different aspects.
3.	Adequate materials are well provided, including the starting code, the extracted visual features, the original annotations, etc.


**Weaknesses:**

1.	The contribution is limited. It seems this work is a kind of integration of other datasets. The integrating process seems to require moderate and routine efforts.
2.	The motivation is not strong enough compared with other works aiming at unifying existing datasets. For example, the AMASS dataset unifies different motion capture datasets because those datasets have different settings (e.g. markers set), which hinders the training of deep models. Those authors have to propose creative methods to perform the integration.
3.	Concerns about different granularities of different datasets. To explain a little, the authors unifying a variety of datasets for different tasks, some of them focused on more general situations (those in TV and movies, etc.) and some limited to more restricted scenes (\e.g. YouCook2). It’s suggested that the unifying process as well as the evaluation should be carried based on dataset granularity.


**Additional Feedback:**

Please consider the semantic and temporal granularity when targeting at unifying datasets.
For example, scenes in TV shows and movies are more general,
while there are some datasets built for a specific domain, e.g. MPII Cooking, FineGym, Epic Kitchen. Please discuss them in related work.

**Clarity:**

The structure is clear but lacks necessary figures. Figures are suitable to give a better overview of the dataset and could display the experiment results visually.

**Correctness:**

Some claims need more evidence to support. For example, in Line 232 “QA tasks benefit more from subtitle channel than video.” This is caused by the choice of datasets, meaning that answering video-related questions from these datasets requires little focus on visual content and the model could cheat using language bias. For some datasets where “vision really counts”, e.g. the FineGym dataset, things may be different.

**Documentation:**

URL for the dataset is provided. Starter code, visual features, subtitles and annotations are released.

**Ethics:**

No.

**Relation To Prior Work:**

This work is an assemble dataset of some previous datasets.
I hope to see more clarification about the main difference except this dataset is large.


**Summary And Contributions:**

This paper proposes a unified dataset for comprehensive video and language understanding.
Particularly, they unify 11 different existing datasets, targeting three cross-modality tasks, namely video retrieval, video question answering, and video captioning.

---

> ### Author Response · Authors · 2021-07-14
> **Thanks for the valuable feedback, we address each of the comments below.**
>
> `Contribution`
>
> (1) We compile a set of diverse video+language datasets/tasks to provide a comprehensive & standardized evaluation to video+language understanding models, similar to GLUE and SuperGLUE benchmarks for NLP. We believe VALUE is a timely and useful resource to foster future works on video+language. See `Motivation` below for more about our standardized efforts.
>
> (2) We conduct in-depth analysis (as noted by the other reviewers) on VALUE tasks to provide modeling insights to future work. As a benchmark paper, despite providing high-quality data, we believe this is also a very important contribution to include.
>
> `Motivation` and `Unifying Datasets with More Creative Methods`
>
> Our core motivation is to provide a unified evaluation to video+language understanding. Prior models in this domain use different video features, work on different tasks/datasets/evaluation splits, different context settings (e.g., video only or video+dialogue) (L73-74). For example, on YC2C, [1] takes pure video as input, while [2,3] leverage both video and transcribed dialogue. This difference makes it hard to fairly compare model performance. Furthermore, even for [2, 3], the transcribed dialogue can be different depending on the ASR models used. To standardize such settings, we provide a single set of transcribed dialogue for all videos. This is in a similar essence to how AMASS unifies different marker sets.
>
> From our understanding, it is easier to unify different datasets in AMASS as they share a common human body representation (markers). However, the tasks and datasets in VALUE are from diverse video domains, of diverse video lengths, annotated in different formats with different task formulations (e.g., retrieval vs. QA). Hence, we try to provide unified video representations leveraging existing vision models. In addition, we leverage the commercial-level ASR model to provide high-quality transcribed dialogues as additional input context for unified multi-channel video+language understanding evaluation, which is in a similar essence to how AMASS unifies different marker sets.
>
> Further, we would like to emphasize that the construction of VALUE is not trivial. For example, (i) The original How2R and How2QA datasets are noisy. Hence, we have spent additional human efforts to recalibrate How2R and How2QA to provide a cleaner version (L119-120); (ii) As mentioned above, we augment datasets with ASR as additional input channel for YouCook2 and VATEX to unify all videos in VALUE as multi-channel videos (L125-126, L133);(L125-126, L133); (iii) We provide previously unreleased test data for TVQA/TVR/TVC/YC2R/YC2C datasets; (iv) We perform human evaluation and report human performances on eligible VALUE tasks (L304 and Appendix D).
>
> As Reviewer sp5p nicely summarizes for us: “Unified evaluation metrics, pre-extracted features, benchmark, starter code, and leaderboard are provided, so the benchmark is well ready to be investigated and make an impact to researchers in this field.”
>
> [1] Zhou, Luowei, et al. "End-to-end dense video captioning with masked transformer." CVPR, 2018.
>
> [2] Hessel, Jack, et al. "A case study on combining asr and visual features for generating instructional video captions." arXiv preprint arXiv:1910.02930 (2019).
>
> [3] Shi, Botian, et al. "Dense procedure captioning in narrated instructional videos." ACL, 2019.
>
> `Different Granularities`
>
> Thanks for your valuable suggestions. We agree that the videos in TV and YouCook2 are indeed from different granularities at the semantic level. Hence, we provide and encourage future work to report evaluation of models at different levels: each dataset, each task, all datasets+all tasks, to ensure a comprehensive evaluation of model performance.
>
> In addition, the differences in video domains suggest that VALUE includes diverse video contents. An ideal video+language understanding model should perform well on both general domains (such as TV/Movie) and more restricted domains (such as Cooking). Together with diverse task types, we believe VALUE can encourage development of such more-general video+language understanding systems.

---

> > ### Author Response · Authors · 2021-07-14
> > **Additional Response to Reviewer's Comments**
> >
> >
> > `General vs. Restricted Domain (e.g., MPII Cooking, FineGym, Epic Kitchen)`
> >
> > From our understanding, the reviewer is asking about related work on datasets from more general domains versus from more restricted domains. The VALUE benchmark is to provide a comprehensive video+language understanding evaluation. Therefore, we try to include diverse datasets/tasks from different domains.  In current literature, there are not many *video+language* datasets that focus on specific domains, like YouCook2 for cooking. Hence, we do not provide an evaluation metric tailored for restricted domains.
> >
> > The datasets mentioned in the reviewer's comment are designed primarily for pure video understanding tasks, which are different from the focus of the VALUE benchmark. For example, Epic Kitchen is created for object detection, action recognition and anticipation tasks, FineGym is designed for fine-grained gymnastic action classification from videos. In contrast, VALUE focuses on both video and language understanding, e.g. retrieval, QA and captioning.
> >
> > `Correctness of L232`
> >
> >  We agree that the claim made in L232 is solely based on the QA datasets included in VALUE, and the conclusion might be different for datasets built on single-channel videos (like FineGym). The goal of our analysis in Section 4 is to help future work better understand the datasets in VALUE and provide insights on future model design choices. It is not intended to provide a general conclusion/observation to all video QA datasets, especially the ones not in VALUE. Meanwhile, the smaller gain from adding visual content does not suggest visual content is not important for the task, it may be caused by the greater difficulties of visual understanding in the task.
> >
> > We also want to reiterate to emphasize that VALUE focuses on video+language datasets where both video and subtitle are provided as context for downstream tasks. In real-life scenarios (e.g., YouTube videos), videos are often accompanied with speech signals and such signals are a crucial part of the video. Therefore, we did not include datasets like FineGym in VALUE.
> >
> > `The structure is clear but lacks necessary figures.`
> >
> > In the updated version, we have moved the dataset/task illustrative figure (Figure 1) from Appendix to the main text. We have also included figures on video length distribution in the Appendix (Figure 2). We will look into adding more figures, for example, more dataset figures and experimental results figures.

---

### Official Review · Reviewer_CWvD · 2021-07-05
**Review for VALUE benchmark**

**Rating:** 8
**Confidence:** 3
**Clarity:** The paper is well-written and presented.

**Strengths:**

The benchmark is well-motivated and the baseline evaluations are well-designed. I particularly welcome the focus on the multi-modal aspect of the benchmark, combining the NLP and vision. Overall, the paper makes a robust contribution to the community.

**Weaknesses:**

.

**Additional Feedback:**

.

**Correctness:**

The experimental setup for understanding the baseline architecture's performance on the benchmark is conducted in a thorough and reasonable manner, testing whether the individual video or the language component of the dataset is enough to achieve a high enough performance, and investigating how different ways to fuse these multimodal components into an embedding. The experiments also point to the potential of joint learning between different tasks, which is interesting in the context of the recent trend to create a single universal model that can perform robustly across multiple tasks.

**Documentation:**

The documentation is clear and the website hosting the dataset and the competition is largely easy to follow and accessible. I also welcome the use of CodaLab for replicability.

**Ethics:**

This benchmark would not incur any ethical concerns beyond what would have been stated in the 11 component datasets.

**Relation To Prior Work:**

The paper adequately cites and identifies a missing link in the current language and video datasets, modeling itself after the recent GLUE and SuperGLUE benchmarks that aim to combine multiple datasets and tasks to evaluate models in a more general setting.

**Summary And Contributions:**

This paper introduces a new multi-task benchmark for video and language understanding tasks. The benchmark is organized in a fashion that is reminiscent of the recent GLUE and Super-GLUE datasets in the NLP community, combining 11 video and language datasets that are diverse, challenging, and hard-to-game. By covering a wide range of tasks, the benchmark aims to measure the effectiveness of a universal video and language system beyond just a single task. Finally, the paper is complemented with a series of experimental results using baseline models that showed that there is a significant gap between the baseline models and the human performance, suggesting that there is ample room for improvement that can be obtained using the benchmark as the guide.

---

> ### Author Response · Authors · 2021-07-14
> **Thanks for the encouraging and constructive comments!**
>
>  We are encouraged that the reviewer finds that the VALUE benchmark can make a robust contribution to the community. We believe the VALUE benchmark can be a timely and useful resource to foster future works on video+language understanding. The diverse datasets and tasks included in VALUE may also inspire advances in developing general video+language understanding systems.
> We also appreciate the reviewer’s recognition that our experiments and analysis are conducted in a thorough and reasonable manner, and can provide insights in future model design choices along this direction.

---

### Decision · Program_Chairs · 2021-07-26

**Decision:**

Accept

**Comment:**

The paper provides an extensive benchmark and thorough evaluation for several video and language understanding tasks. The contribution has the potential to accelerate future research.